# Novel aerosol treatment of airway hyper-reactivity and inflammation in a murine model of asthma with a soluble epoxide hydrolase inhibitor

Chuanzhen Zhang[1,2], Wei Li[3]*, Xiyuan Li[2,3], Debin Wan[4], Savannah Mack[2], Jingjing Zhang[2], Karen Wagner[4], Chang Wang[4], Bowen Tan[4], Jason Chen[2], Ching-Wen Wu[2], Kaori Tsuji[5], Minoru Takeuchi[5], Ziping Chen[1], Bruce D. Hammock[4], Kent E. Pinkerton[2]*, Jun Yang[4]*

1 Department of Gastroenterology, the First Affiliated Hospital of Shandong First Medical University & Shandong Provincial Qianfoshan Hospital, Jinan, Shandong, China, 2 Center for Health and the Environment, University of California, Davis, California, United States of America, 3 School of Control Science and Engineering, Shandong University, Jinan, Shandong, China, 4 Department of Entomology and Nematology and Comprehensive Cancer Center, University of California, Davis, California, United States of America, 5 Department of Animal Medical Science, Kyoto Sangyo University, Kyoto, Japan

* cindy@sdu.edu.cn (WL); kepinkerton@ucdavis.edu (KEP); junyang@ucdavis.edu (JY)

**Data Availability Statement:** All relevant data are within the paper and its Supporting Information files.

## Abstract

Asthma currently affects more than 339 million people worldwide. In the present preliminary study, we examined the efficacy of a new, inhalable soluble epoxide hydrolase inhibitor (sEHI), 1-trifluoromethoxyphenyl-3-(1-propionylpiperidin-4-yl) urea (TPPU), to attenuate airway inflammation, mucin secretion, and hyper-responsiveness (AHR) in an ovalbumin (OVA)-sensitized murine model. Male BALB/c mice were divided into phosphate-buffered saline (PBS), OVA, and OVA+TPPU (2- or 6-h) exposure groups. On days 0 and 14, the mice were administered PBS or sensitized to OVA in PBS. From days 26–38, seven challenge exposures were performed with 30 min inhalation of filtered air or OVA alone. In the OVA+TPPU groups, a 2- or 6-h TPPU inhalation preceded each 30-min OVA exposure. On day 39, pulmonary function tests (PFTs) were performed, and biological samples were collected. Lung tissues were used to semi-quantitatively evaluate the severity of inflammation and airway constriction and the volume of stored intracellular mucosubstances. Bronchoalveolar lavage (BAL) and blood samples were used to analyze regulatory lipid mediator profiles. Significantly ($p < 0.05$) attenuated alveolar, bronchiolar, and pleural inflammation; airway resistance and constriction; mucosubstance volume; and inflammatory lipid mediator levels were observed with OVA+TPPU relative to OVA alone. Cumulative findings indicated TPPU inhalation effectively inhibited inflammation, suppressed AHR, and prevented mucosubstance accumulation in the murine asthmatic model. Future studies should determine the pharmacokinetics (i.e., absorption, distribution, metabolism, and excretion) and pharmacodynamics (i.e., concentration/dose responses) of inhaled TPPU to explore its potential as an asthma-preventative or -rescue treatment.

**Funding:** This work was supported in part by grants from the American Asthma Foundation 15-Hamm-Ex1, NIEHS/Superfund Research Program P42 (ES004699), NIH/NIEHS R35 (ES030443), National Institute for Occupational Safety and Health (OHO7550), and the National Institute of Environmental Health Sciences P30 (ES023513) and P51 (OD011107). The funders had no role in study design, data collection and analysis, decision to publish, or preparation of the manuscript.

**Competing interests:** B.D. Hammock and J. Yang are co-authors on University of California patents for epoxide hydrolase inhibitors (sEHIs). An sEHI which is structurally similar to the TPPU used in this study, is being developed by EicOsis Human Health to replace opioid analgesics in treating pain. Drs. Hammock and Yang are part-time employees of EicOsis Human Health. This does not alter our adherence to PLoS One policies on sharing data and materials. Data in this publication were not generated from grant- or salary-money from EicOsis. This research has no financial connections to EicOsis. No confidential University information has been provided to EicOsis by the University of California or its employees. This does not alter our adherence to PLOS ONE policies on sharing data and materials.

## Introduction

Asthma is one of the most common and costly chronic respiratory disorders worldwide [1]. According to the 2018 *Global Asthma Report* [2], asthma affects approximately 339 million people worldwide, with an additional 100 million people projected to be impacted by 2025. Asthma is also a consequence of complex gene-environment interactions involving factors including but not limited to allergies, stress, air pollution, tobacco use, occupational risk, and microbial exposure [3]. The pathogenesis of asthma is compounded by airway inflammation, airway hyper-responsiveness (AHR), and airway remodeling [4]. Clinically, asthma symptoms (e.g., wheezing, dyspnea, coughing, and chest tightness) are variable and non-specific, manifesting as chronic, episodic, and reversible attacks.

National and international asthma-management guidelines exist [1, 5, 6]. The most successful asthma treatments are personalized to include non-pharmacological interventions such as self-management education and treatments for risk factors and comorbidities, as well as asthma-targeted medications. Beta (β)-2 agonists, anticholinergics, corticosteroids (parenteral, oral, or inhaled), epinephrine, magnesium sulfate, heliox-driven albuterol, leukotriene (LT) receptor antagonists, and immunosuppressants are among the traditional types of asthma medications [7]. New biologics have recently been developed to target effector cells, cytokines, and molecules to inhibit eosinophils, interleukin (IL)-5, IL-9, immunoglobulin E, and epithelial-derived cytokines involved in asthma [8]. These novel compounds have proven efficacy in targeting specific drivers of asthma-related symptoms. However, healthcare access, substandard care provider-patient communication, medication issues, comorbidities, and mood disorders hinder asthma treatments [9].

Despite numerous effective asthma medications, there remains a need for novel drugs to prevent or alleviate asthmatic symptoms and maintain physiological function with minimal adverse side effects to deter patient compliance. Soluble epoxide hydrolase inhibitors (sEHIs) are currently being tested as novel treatments for inflammation, pain, and other disease states [10–12], with fewer side effects than nonsteroidal anti-inflammatory drugs. The pro-inflammatory soluble epoxide hydrolase (sEH) enzyme hydrolyzes biologically active, anti-inflammatory, and inflammation-resolving epoxy fatty acids (EpFAs) to less bioactive diols [10, 12]. EpFAs are the lipid mediators derived from polyunsaturated fatty acids (PUFAs) via enzymatic cytochrome P450 (CYP450) reactions. EpFAs play fundamental roles in preventing inflammation, relieving pain, improving degenerative diseases [12], and promoting vasodilation, anti-hypertrophy [13], and angiogenesis processes [14]. Epoxyeicosatrienoic acids (EETs, also known as EpETrEs), a common class of EpFAs, have been shown to inhibit the release of pro-inflammatory immune mediators (e.g., histamine) by human lung mast cells activated in hyper-reactive airways [15]. One example is 14,15-EET, which produced beneficial biological outcomes, including suppressing airway remodeling in a chronic asthma model [16, 17]. It should be noted that linoleic acid (LA), the most abundant PUFA in the Western diet [18], is converted into epoxyoctadecenoic acids (EpOMEs) through the CYP450 pathway and metabolized by sEH into dihydroxyoctadecenoic acids (DiHOMEs). Among the EpOMEs, 9,10-EpOME and 12,13-EpOME, known as leukotoxin and isoleukotoxin, respectively, are involved in inflammatory neutrophil respiratory bursts [19]. EpOMEs and DiHOMEs have been implicated in chronic pulmonary diseases, including asthma, by mediating inflammation and immune responses [20].

Like CYP450 metabolism, the cyclooxygenase (COX) and lipoxygenase (LOX) pathways (S1 Fig) also result in the formation of antiphlogistic mediators via PUFA metabolism. One class of metabolites from the LOX pathway includes anti-inflammatory lipoxins (LXs), which are increased upon sEHI administration in severe asthma patients [21].

Asthma severity has been linked to imbalances in the levels of LTs relative to LXs, both of which are products of lipoxygenase-derived eicosanoid biosynthesis. For example, higher mean levels of pro-inflammatory LTs and lower basal levels of LXs have been reported in patients with severe versus moderate asthma [22]. Several studies show that LXs are generally beneficial, promoting the resolution of allergic inflammation [23]. In a separate study [21], we showed that LX generation is related to sEH activity in severe asthmatic patients. LXs may regulate allergic airway responses (e.g., inflammation, mucus metaplasia, and AHR) by inhibiting eosinophil trafficking, neutrophil chemotaxis, and degranulation [21]. The CYP450, COX, and LOX pathways are cross-linked in various physiological responses related to inflammation and other diseases [24, 25].

Given that sEH is 1) well-conserved among different species [26], 2) found in normal and inflamed human lung tissues, and 3) detected at higher levels along with its metabolic products in severe allergic airways relative to normal airways [27], sEHIs have been increasingly researched as potential treatments for asthma and other respiratory diseases. Previous studies with different sEHIs have demonstrated their abilities to reduce respiratory airflow resistance [28], suppress allergic airway inflammation and AHR [24], and inhibit airway remodeling [16] and pulmonary fibrosis [29] in rodent models of allergy, asthma, and chronic obstructive pulmonary disease. Preliminary investigations (unpublished data) suggest that sEHIs synergize with the 5-lipoxygenase activating protein (FLAP) inhibitor, MK886, to reduce airway inflammation in the ovalbumin (OVA)-induced murine model. MK886 inhibits the biosynthesis of inflammatory LTs [30] that cause bronchoconstriction and excess mucus formation inherent to asthma [31]. Several sEHIs have been developed for the treatment of hypertension, Type-2 diabetes [10, 11], and chronic pain [12] and subsequently moved into human clinical trials [32].

Over the last few years, one sEHI, 1-(4-trifluoro-methoxy-phenyl)-3-(1-propionylpiperidin-4-yl) urea (TPPU), has been used in numerous studies to evaluate the role of sEH in disease states. TPPU has not been shown to bind to other (non-sEH) targets. However, TPPU was found to exhibit 1) high inhibition potency without apparent non-specific binding [33]; 2) favorable solubility and pharmacokinetics; and 3) biological activity that has been well-characterized as it pertains to metabolism [34]. Though its analog, EC5026, was recently studied in healthy human subjects as an oral analgesic [35], TPPU has never been tested in clinical trials. Additionally, despite that the physiochemical properties of TPPU enable its administration via inhalation to directly target the pathological sites of asthma, there are currently no reports of TPPU being tested via this route. Given these factors, the needs unmet by current asthma medications, the potential relevance of TPPU to the full spectrum of asthma disease phenotypes, and the favorable therapeutic window of TPPU, we used a well-established murine asthmatic model in our preliminary TPPU experiments. These experiments were performed to test the hypothesis that inhaled TPPU prevents asthma by reducing airway inflammation, hyper-responsiveness, and remodeling.

## Materials and methods

### Animals, exposures, and treatments

All animal experiments were approved in writing by the University of California Davis' Institutional Animal Care and Use Committee (IACUC) to ensure the proper care and treatment of animals in our research.

Twenty-six 8-week old male BALB/c mice (Envigo, Hayward, CA) were purchased and randomly assigned to four separate, weight-matched exposure groups. All mice were housed four/cage, allowed to acclimate for one week before the study began, and provided food and water

*ad libitum* except during the inhalation exposures. Each mouse served as an experimental unit in the present study. The group sizes were calculated to ensure sufficient power to determine statistically significant inter-group differences, as described in a previous study [24].

The mice were divided into phosphate-buffered saline (PBS) negative control (n = 8), ovalbumin (OVA; 1%) positive control (n = 6), and OVA+TPPU (2- or 6-h; n = 6 each) groups. OVA is an allergenic protein derived from chicken egg whites and commonly used to produce murine models of acute asthma, with reproducible airway inflammation, hyper-responsiveness, and remodeling [36]. The PBS and lyophilized OVA powder were obtained from Sigma Aldrich (St. Louis, MO), and the OVA powder was dissolved in the PBS to produce the 1% OVA used in the present study. The TPPU was synthesized in the laboratory of Prof. Bruce Hammock (UC Davis, USA) [33], dissolved in polyethylene glycol 400 (PEG400, Sigma Aldrich) at 100x concentrates, and diluted in distilled water before administration.

The animals were treated following a previously published protocol [24] with some modifications. Briefly, on days 0 and 14, the mice were administered PBS (PBS control group only) or sensitized to 10 mL of 1% OVA via intraperitoneal (IP) injection. From days 26–38, 30-min inhalation challenge exposures were performed every other day for seven total exposures to filtered air (PBS control group only) or nebulized 1% OVA alone. In the OVA+TPPU groups, a 2- or 6-h TPPU (1 mg/mL) inhalation treatment preceded each 30-min OVA exposure.

The 1 mg/mL TPPU concentration was based upon studies [24, 37] of sEHIs in asthma, acute lung injury, and other disease models. The TPPU aerosol was generated using a Mini-Heart Lo-Flo Nebulizer (Westmed Inc., Tucson, AZ) attached to a nose-only exposure chamber. The aerosol flowed through the chamber at 2 liters/min at 30 psi. During the time of exposure, each mouse was housed in a cylindrical nose-only exposure tube (Teague Enterprises, Woodland, California). The tubes were connected to the exposure system for the TPPU aerosol exposures or placed on a table in the same room as the exposure system for the filtered air (control) exposures.

On day 39, pulmonary function tests (PFTs) and necropsies were performed on all mice. Bronchoalveolar lavage (BAL) and blood samples were collected to analyze regulatory lipid mediator profiles. Lung tissues were analyzed semi-quantitatively to determine the severity of inflammation and airway constriction and volume of stored intracellular mucosubstances. The complete exposure paradigm is illustrated in Fig 1. All tests, analyses, and semi-quantitative evaluations were performed blinded.

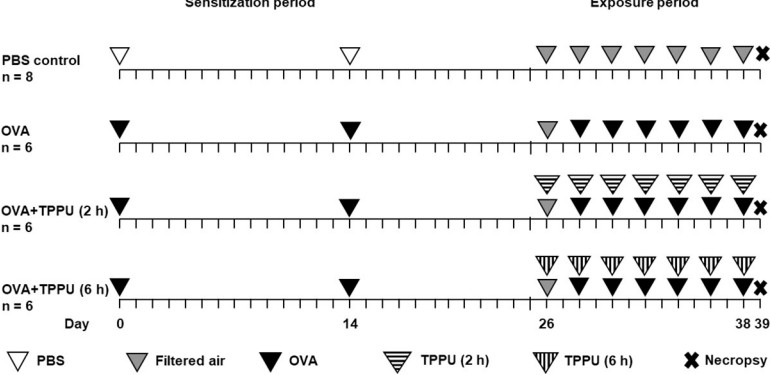

**Fig 1. Diagram of the study design and timeline.** On days 0 and 14, the mice were administered PBS (PBS control group only) or sensitized to 1% OVA via intraperitoneal injection. From days 26–38, 30-min inhalation challenge exposures were performed every other day for a total of seven exposures to filtered air (PBS control group only) or nebulized 1% OVA alone. In OVA+TPPU groups, a 2- or 6-h TPPU (1 mg/mL) inhalation treatment preceded each 30-min OVA exposure. The 1-mg/mL TPPU concentration was based upon studies of sEHIs in asthma, acute lung injury, and other disease models. On day 39, pulmonary function tests and necropsies were performed on all mice.

## Lung physiological measurements

On day 39, the mice were weighed and anesthetized with an IP injection of tiletaminezolaze-pam (50 mg/kg) and dexmedetomidine (0.7 mg/kg) and an intramuscular injection of succi-nylcholine (75 mg/kg). Each animal was intratracheally cannulated for PFT and AHR measurements using a Flexivent system (SCIREQ, Montreal, Canada). As AHR measurements are determinants of asthma severity, sequential aerosol challenges with increasing methacho-line (MCh) concentrations (0.0625, 0.125, 0.25, and 0.5 mg/mL) were used to gradually increase the severity of airway constriction and measure resistance in the mice. The methacho-line challenges provided a means to evaluate the effects of different treatments (i.e., PBS, OVA, and OVA+TPPU) in the present study. For example, a comparatively increased airway resis-tance at a lower versus higher MCh concentration would indicate a hyper-reactive airway response. The maximum tested MCh concentration was limited to 0.5 mg/mL because OVA-exposed mice exhibited severe dyspnea and could not tolerate higher concentrations found in preliminary studies [38, 39].

Ventilation was set with a 10-mL/kg tidal volume at 200 breaths/min. After nebulization of 1 mL PBS or MCh for approximately 2 min, airway resistance was monitored for an additional 6 min to allow the full aliquot to run through the Flexivent system. Resistance was measured as an index of airflow influenced by airway secretions and/or constriction. This method used input impedance to simulate airway resistance in the peripheral lungs and oscillating perturba-tion [40] to analyze the collected data. Each subject datum point reflected an average of responses measured in triplicate after each PBS or MCh exposure challenge.

## Necropsy and lung fixation

Immediately following the completion of physiological measurements and before collection of blood, bronchoalveolar lavage fluid (BALF), and lung tissues, the mice were euthanized with an IP injection of beuthanasia-D (3 mL; 1 mg/mL). For each mouse, blood was collected by cardiac puncture, and the plasma was separated for analysis of sEHI levels and lipidomic pro-files as described in the section, "TPPU concentration measurement and regulatory lipid medi-ator profiling." The left lung was clamped, while the right lung was perfused with two sequential aliquots (0.7 and 0.6 mL) of sterile PBS to collect BALF samples. The two BALF samples from each mouse were pooled, and the supernatant was prepared for liquid chroma-tography. The right lung tissues were flash-frozen at -80°C, while the left lung was inflation-fixed for histology.

## Lung histopathology

The left lung of each mouse was inflation-fixed with 4% paraformaldehyde (Electron Micros-copy Sciences, Hatfield, Pennsylvania) under 30 cm of water pressure for 1 h, stored in fixative for 24 h, and subsequently transferred into 70% ethanol. The left lung lobe was prepared as four transverse slices corresponding to different axial levels from the cranial to caudal portions of the lung. All lung tissues were dehydrated through a graded series of ethanol and toluene and embedded in paraffin (Paraplast X-tra, Leica, Richmond, IL). Tissue sections were pre-pared to a thickness of 5 μm using a microtome and mounted onto glass slides. Each slide con-tained all four lung tissue levels from a mouse. One slide was stained with hematoxylin and eosin (H&E; American Mater Tech Scientific, Inc., Lodi, CA, USA). The second slide was stained with alcian blue/periodic acid-Schiff (AB/PAS; American Mater Tech Scientific, Inc., Lodi, CA, USA). An Axiolab light microscope (Zeiss, Jena, Germany) was used to view each stained lung section. Zeiss ZEN 2.5 software was applied to project the microscope view taken

by a Zeiss Axiocam 105 color camera (Zeiss, Jena, Germany) onto a desktop computer for high-resolution image capture in the Carl Zeiss and TFI imaging formats.

H&E-stained slide images were used for semi-quantitative analysis of inflammation for all lung tissue sections. Bronchiolar, alveolar, perivascular, and pleural inflammation were scored using an ordinal scale. Severity scores ranged from 0 to 3, corresponding to no, mild, moderate, and marked cellularity of inflammation, respectively (S1 Table). The extent of inflammation was defined as the distribution of cellular change across the lung tissue sections for each mouse, with scores of 0 (no inflammation of the lung), 1 ($\leq$ 1/3 tissue involvement), 2 (1/2 tissue involvement), and 3 ($\geq$ 2/3 tissue involvement). Severity and extent scores for each mouse were multiplied and averaged for each anatomical region to define the final bronchiolar, alveolar, perivascular, and pleural inflammation scores. This method was described previously [41] and modified slightly for the present study to describe airway inflammation changes more fully.

**Severity of airway constriction.** H&E slide images were magnified using 5x to 20x objective lenses to measure airway constriction. Airway constriction was quantified separately for each lung level using two parameters. These parameters included the severity of airway constriction, which was scored using a pre-defined ordinal scale (S2 Table), and the ratio of constricted airways to total airways (constricted airways/total airways). These two values were multiplied to determine an airway constriction score for each lung level. The resulting four lung level scores were averaged to assign an overall airway constriction score for each mouse.

**Airway intracellular mucosubstance measurement.** AB/PAS-stained slides were used to measure the abundance of mucosubstances in the airways. Each lung level was analyzed separately, but all airways for all four levels were digitally captured using a 40x objective lens, and the basal lamina in each airway was digitally outlined by hand. Mucin production was quantified using ImageJ software (version 2.0.0; National Institutes of Health, Bethesda, MD, USA). The volume of intracellular mucosubstances per basal lamina surface area (quotient) was calculated using the ImageJ software color threshold function, which enabled quantification of the intracytoplasmic mucin area per basal lamina length. An average of measurements for all four levels was calculated to determine airway mucosubstance abundance for each mouse.

## TPPU concentration measurement and regulatory lipid mediator profiling

In asthma, sEHIs preserve anti-inflammatory EpFAs and reduce pro-inflammatory diols by inhibiting sEH [12]. Thus, we assessed the ability of inhaled TPPU to engage sEH by measuring concentration changes in the *in vivo* substrates (EpFAs) and products (diols) of the sEH enzyme and other lipid mediators from the adjacent COX and LOX pathways (S1 Fig). TPPU quantification and regulatory lipid mediator profiling by ultra-high-pressure liquid chromatography-tandem mass spectrometer (UHPLC MS/MS) were performed using an Agilent 1200 SL (Agilent, Santa Clara, CA, USA) liquid chromatograph coupled to an AB Sciex 4000 QTrap (AB Sciex, Redwood City, CA, USA) MS/MS system in which the mass spectrometer was operated under negative electrospray mode [42].

For each mouse, 10 μL of blood plasma was aliquoted into 50 μL of ethylenediaminetetraacetic acid (EDTA) water solution to quantify the concentrations of TPPU. Before analyzing the TPPU concentration, the plasma-EDTA solution was extracted with 200 μL of ethyl acetate twice after spiking with 10 μL of a 500-ng/mL deuterated isotope-labeled TPPU (d5 TPPU) solution. The extracts were dried using a vacuum concentrator (Neutec, Farmingdale, NY, USA) for approximately 30 min at 70 mbar and reconstituted with 50 μL of methanol solution before analyzing by a UHPLC MS/MS that was developed for the quantification of TPPU in complex matrices. The mobile phases in the LC analysis of blood plasma were 0.1% acetic acid

in a water solution (mobile phase A) and 0.1% acetonitrile solution (mobile phase B). The liquid chromatography gradient started from 35% of B and increased to 75% of B in 2 min at a 0.4-mL/min flow rate. The column used to quantify TPPU was a Kinetex C18 reverse-phase column (50 x 2.1 mm, 1.7 um particle size) from Phenomenex (Torrance, CA). The parameters were optimized using an authentic TPPU standard synthesized in the lab.

To determine lipidomic profiles of 88 targeted metabolites derived from PUFAs—including prostaglandins (PGs), LTs, EpFAs, and others—the collected BALF supernatant and plasma samples were extracted following previously published protocols [24, 34]. Briefly, the samples (BALF supernatant or plasma) were loaded onto a pre-washed solid-phase extraction cartridge (Oasis HLB 3-cc solid-phase extraction cartridge, Waters, Milford MA) after spiking an internal standard mixture that included ten isotope-labeled lipid mediators. Then, the cartridge contents were eluted with 0.5 mL of methanol and 1.5 mL of ethyl acetate sequentially. The methanol and ethyl acetate eluents were combined and concentrated before being injected onto columns. The UHPLC-MS/MS program parameters were optimized using authentic standards of the measured lipid mediators [32]. The analyses were conducted as previously described [34]. The 88 lipid mediators were measured at a 1-nmol/L limit of quantification in 21.5 min using this optimized protocol.

## Statistical analysis

All data were analyzed and graphed using GraphPad Prism 8 software (GraphPad, Inc., San Diego, CA, USA). Shapiro-Wilk and Bartlett's tests were performed to determine whether the data residuals met the normal distribution and equal variance (homoscedasticity) requirements, respectively, for an analysis of variance (ANOVA). One-way ANOVAs and post-hoc Tukey's Multiple Comparisons tests were used only for normally distributed, homoscedastic data sets. Data sets that failed to pass the Shapiro-Wilk test (i.e., those with a $p$-value < 0.05) were assessed with a Kruskal-Wallis test and post-hoc Dunnett's Multiple Comparisons test rather than an ANOVA. Multivariate models were implemented for lipidomic analyses using MetaboAnalyst 4.0 (freely available at http://metaboanalyst.ca) [43]. Findings with a $p$-value < 0.05 were considered statistically significant. Data were presented as arithmetic group means ± standard errors of the means (SEMs).

## Results and discussion

### TPPU inhalation influenced the body weights of mice relative to OVA alone

Before euthanasia, the average group body weight was 26.6 ± 0.4 g for PBS, 28.2 ± 0.5 g for OVA, 25.5 ± 0.7 g for OVA+TPPU (2 h), and 24.6 ± 0.3 g for OVA+TPPU (6 h). While the difference between the OVA and PBS groups was not statistically significant ($p$ = 0.058), the mean weight of the OVA group was significantly greater than those of the two TPPU treatment groups ($p$ = 0.0039 for 2 h, and $p$ < 0.0001 for 6 h).

Previous studies have reported increased lung weights with [44] or without [36] concomitant increases in body weights after OVA exposure. Lung weights were not measured in the present study. Although edematous inflammation and mucin hypersecretion in the lungs of OVA-exposed mice [45] may have contributed to increased body weight relative to the other mice, the differences in body weight were more likely due to varying levels of activity between treatment groups. Asthmatic mice are often reluctant to exercise because vigorous exercise could trigger bronchoconstriction in their relatively hypoxic state [46]. Therefore, in the present study, it is plausible that TPPU-treated mice had higher activity levels than their asthmatic

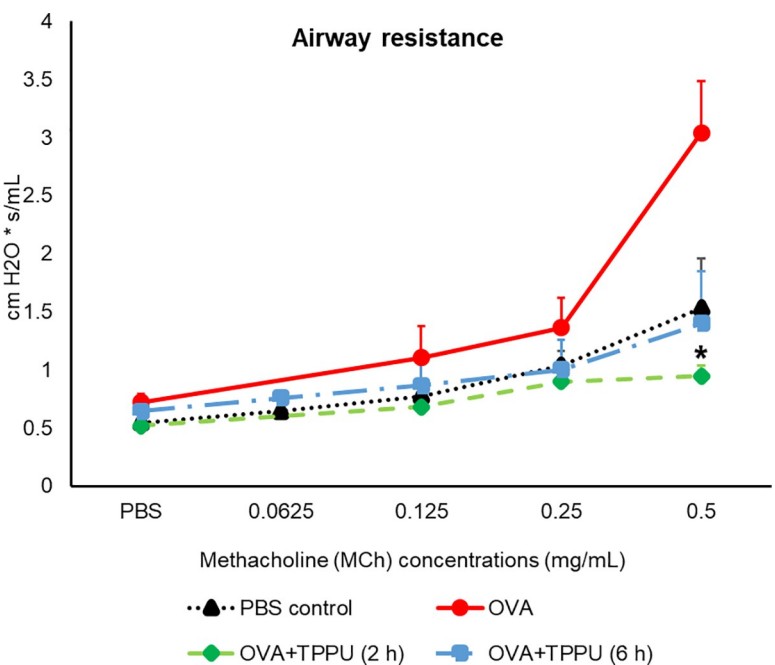

**Fig 2. Airway resistance in MCh-challenged mice.** All data are shown as the mean ± SEM (n = 5–8). An asterisk (*) represents a significant difference ($p < 0.05$) between the OVA and OVA + TPPU (2 h) treatments following the 0.5-mg/mL MCh challenge, according to a Kruskal-Wallis test and post-hoc Dunnett's Multiple Comparisons Test.

counterparts given OVA alone. Future studies should confirm whether weight loss is a side-effect of TPPU by measuring body weights throughout the study.

## TPPU limited AHR and airway constriction in OVA-sensitized mice

The PFTs performed using sequentially increasing concentrations of MCh demonstrated OVA alone increased airway resistance compared to the PBS controls (Fig 2). A statistically significant ($p = 0.0201$) difference was observed between the OVA and OVA+TPPU (2 h) groups at 0.5 mg/mL. Although the loss of one mouse in the OVA+TPPU (6 h) group occurred before testing, airway resistance was typically similar in the TPPU-exposed mice and PBS controls, with increasing MCh concentrations.

Similar patterns were observed in semi-quantitative histopathological assessments of airway constriction, with a three-fold lesser degree of bronchiolar constriction following the 0.5-mg/mL MCh challenge in OVA+TPPU (2 and 6 h) mice versus their OVA counterparts ($p < 0.0001$ for both comparisons; Fig 3A and 3B), and no statistical differences between OVA +TPPU- and PBS-exposed mice ($p > 0.05$; Fig 3B).

## TPPU significantly suppressed lung histopathology in OVA-sensitized mice

The robust inflammatory responses produced by OVA alone indicated severe disease. Findings observed in the OVA, but not PBS or OVA+TPPU exposure groups included narrowed and almost occluded airways in Figs 3A (left column) and 4A (right column), cellular debris in the bronchiolar lumen, dense focal accumulations of eosinophils and other inflammatory cells in the alveolar lumen and peribronchiolar regions, and thickening of the bronchiolar epithelium of the lungs (Figs 3A and 4A). TPPU-treated animals exhibited statistically significant

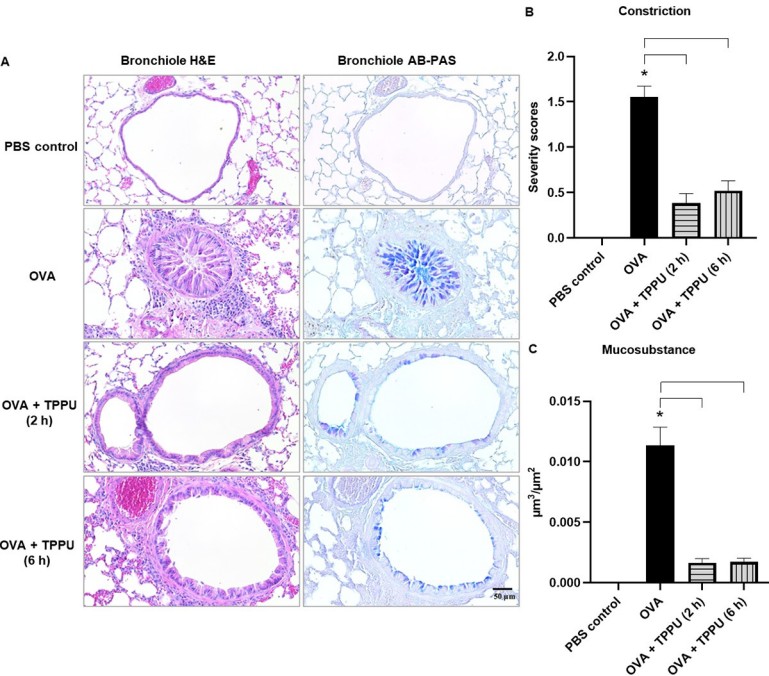

**Fig 3. Histopathology and semi-quantitative scoring of the airway constriction and average intracellular mucosubstance volume in mouse lung tissues.** Micrographs in the left and right columns of Panel A show serially sectioned H&E and AB/PAS-stained lung tissues, respectively. Darker blue staining in the AB/PAS-stained lung tissues indicates intraepithelial mucosubstances. Graphs in Panels B and C show the severity of bronchoconstriction and the measure of mucosubstance production, respectively. All data are shown as the mean ± SEM (n = 6–8). An asterisk (*) denotes a significant difference compared to the PBS control ($p < 0.05$), while brackets represent significant ($p < 0.05$) differences between the OVA and OVA + TPPU groups, according to a one-way ANOVA and post hoc Tukey's Multiple Comparisons Test.

protective effects against OVA-induced lung histopathology. For example, OVA+TPPU (2 h), produced significantly lower alveolar and bronchiolar inflammation scores relative to OVA alone ($p = 0.0100$ and $p = 0.0017$, respectively), as did OVA+TPPU (6 h) versus OVA alone ($p = 0.0037$ and $p = 0.0085$, respectively; Fig 4B and 4C). The alveolitis and bronchiolitis scores in the two OVA+TPPU groups were on par with the non-asthmatic PBS controls ($p > 0.05$; Fig 4B and 4C). OVA+TPPU exposure also produced significantly less pleural inflammation than the OVA treatment ($p = 0.0183$ for 2 h only; S2B Fig). No such exposure-related differences were observed for perivascular inflammation ($p > 0.05$; S2A Fig).

Given AHR is related to the increased pulmonary recruitment of eosinophils, it was not surprising that some of the general patterns of inflammation observed in the bronchoconstriction analysis were reflected in lung tissues scored for histopathology. This was particularly true for the more severe impacts of OVA alone (relative to other treatments), highlighting the mitigative effects of inhaled TPPU on OVA-sensitized mice.

## TPPU inhibited mucus production in OVA-sensitized mice

In the present study, the average volume of intraepithelial mucosubstances per surface area was measured to estimate the degree of airway remodeling. Semi-quantitative analysis of the AB-PAS-stained lung tissues revealed intracytoplasmic mucin within airway epithelial cells of all groups except the PBS controls (Fig 3A, right column). OVA exposure provoked dramatic differences in mucin secretion that corresponded to the airway constriction findings and

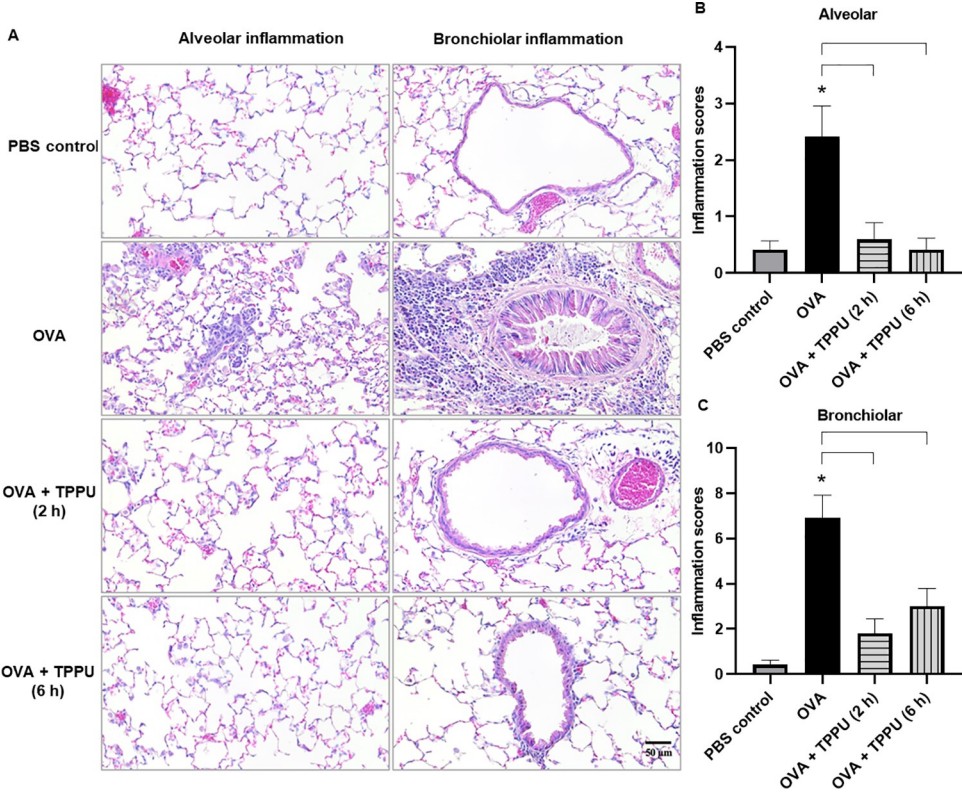

**Fig 4. Histopathological images and semi-quantitative scoring of alveolar and bronchiolar inflammation in mouse lung tissues.** The micrographs in Panel A show serially sectioned H&E-stained lung tissues. The bar graphs in Panels B and C show the mean inflammation scores ± SEMs (n = 6–8). For all three panels, alveolar inflammation is shown on the left, and bronchiolar inflammation is shown on the right. An asterisk (*) denotes a significant ($p < 0.05$) difference compared to the PBS control, while the brackets represent significant ($p < 0.05$) differences between the OVA and OVA + TPPU groups using a one-way ANOVA and post hoc Tukey's Multiple Comparisons Test.

indicated significantly greater remodeling than PBS ($p < 0.0001$; Fig 3C). Mucin secretion in the OVA group was also approximately 6-fold greater than in the OVA+TPPU (2 h) and OVA +TPPU (6 h) groups ($p < 0.0001$ for both; Fig 3C). The two TPPU-treated groups had volumes of intraepithelial mucosubstances similar to the PBS control mice (Fig 3C).

Asthmatic airway remodeling involves the epithelium, smooth muscle, and extracellular matrix components, and has been considered a result of inflammatory insults to the bronchial tissues [47]. Because mucus is the most obvious secretory product of the airway epithelium, excessive mucus production is a hallmark of airway epithelial remodeling [48]. The dramatic differences in mucin secretions of the OVA + TPPU- versus OVA-exposed mice (Fig 3A right column and 3C) indicated that inhaled TPPU could suppress epithelial remodeling of airways in an asthmatic murine model.

## TPPU preserved EpFAs, prevented diol formation, and altered lipid mediators from the COX and LOX pathways in blood and BALF of OVA-sensitized mice

**Effects of TPPU on CYP450-related plasma lipid mediators.** Airway inflammation, airway contraction/dilation [15], and remodeling [17, 48, 49] processes are strongly impacted by EpFAs and diols in the CYP450 pathway. EpFAs have been shown to inhibit the expression of

inflammatory cytokines (e.g., vascular cell adhesion molecule 1), secretion and adherence of tumor necrosis factor-alpha from monocytes, and nuclear translocation of the nuclear factor kappa-light-chain-enhancer of activated B cells [12], which are closely associated with inflammatory responses, AHR, or airway remodeling via various pathways in asthma [50]. In previous studies, EETs were observed stabilizing human mast cells activated in hyper-reactive airways [15] and relaxing airway smooth muscle by activating big calcium-dependent potassium conducting channels and ATP-sensitive potassium channels [51, 52]. Jiang *et al.* [17] recently described the biological anti-remodeling effects of 14,15-EET in a chronic asthma model. In contrast, dihydroxyeicosatrienoic acids (DiHETrEs) have been shown to be influential in monocyte chemoattractant protein-1-induced monocyte chemotaxis, an essential occurrence in inflammation processes [53]. Additionally, 9,10-EpOME was demonstrated to induce pulmonary cell damage or vasoconstriction, followed by nitric oxide-mediated vasodilation [20]. The diol of 9,10-EpOME (i.e., 9,10-DiHOME) could compromise mitochondrial inner membrane permeability and disrupt mitochondrial function [54]. While the 9,10-EpOME and 9,10-DiHOME fatty acids mediate inflammation through activated nuclear factor Kapper-B (NF-κB) and activator protein-1 (AP-1) transcription factors [55], the inflammatory effects of 9,10-DiHOME could be limited by its negative feedback [20]. A murine study [56] recently demonstrated that 12,13-DiHOME could decrease the number of regulatory T cells, cause adaptive immune cell dysfunction, and elevate the risk of developing asthma while increasing pulmonary inflammation.

TPPU blood concentrations for this study were $16.0 \pm 3.4$ ng/mL and $68.6 \pm 12.4$ ng/mL for the 2 h and 6 h inhalation groups, respectively (Fig 5A). These concentrations were nearly 40- and 180-times higher than the half-maximal inhibitory concentration ($IC_{50} = 0.39$ ng/mL) for TPPU in mouse sEH enzymes [34]. The $IC_{50}$ measures the effectiveness of a substance to inhibit a specific biochemical target. In general, a concentration 10-fold higher than the $IC_{50}$ is considered effective. The local concentration in pulmonary cells is likely higher than blood concentrations.

According to the heatmap in Fig 5C, the concentrations of anti-inflammatory EpFAs [e.g., 9(10)- epoxyoctadecadienoic acid (EpODE); 8(9)-EET, 7(8)- epoxydocosapentaenoic acid (EpDPE); 17(18)- epoxyeicosatetraenoic acid (EpETE); and 9(10)- EpOME] were most elevated in the OVA+TPPU (6 h) group (yellow frame in Fig 5C) versus the OVA group. Decreased plasma diol [e.g., 15,16- dihydroxyoctadecadienoic acid (DiHODE); 14,15- dihydroxyeicosatrienoic acid (DiHETrE); 19,20- dihydroxydocosapentaenoic acid (DiHDPE); 14,15- dihydroxyeicosatetraenoic acid (DiHETE); and 12,13- DiHOME] concentrations observed with OVA+TPPU (2 and 6 h) treatments were most similar to the concentrations measured in mice exposed only to PBS, as opposed to OVA alone (white frame in Fig 5C). The observed plasma TPPU concentrations were sufficient to produce a 60% reduction in the plasma concentration of the EpFA diol, 14,15- DiHETrE, in both OVA+TPPU groups compared to mice administered OVA alone ($p = 0.024$ for 2 h, and $p = 0.010$ for 6 h, respectively; Fig 5B).

The plasma TPPU levels observed in the present study strongly suggested inhaled TPPU bound and successfully modulated the action of sEH *in vivo* to cause a systematic change in the sEH pathway rather than a change limited to a few lipid mediators. This systematic change resulted in higher concentrations of anti-inflammatory EpFAs and lower concentrations of diol products (Fig 5) related to asthma [15, 53, 57], such that plasma lipid profiles in TPPU-treated OVA-sensitized mice most resembled those of non-allergic controls. These findings are supported by other studies in which TPPU exposure increased endogenous 14,15-EET in plasma and attenuated lung tissue remodeling in a bleomycin-induced pulmonary fibrosis mouse model [29] and suppressed inflammatory cell influx (total cells, macrophages, and

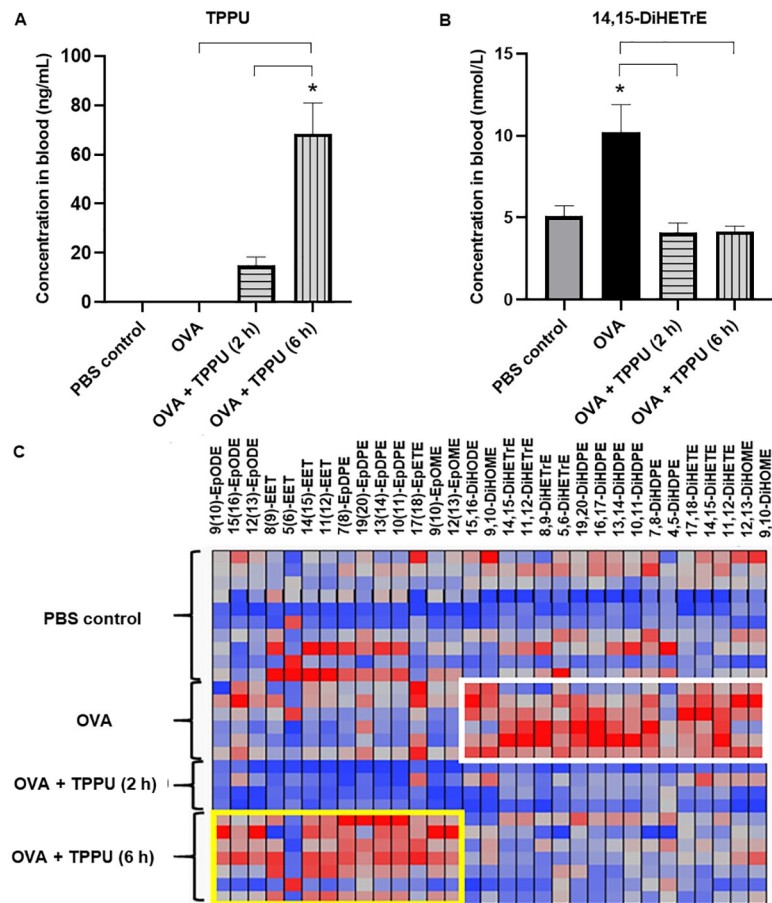

**Fig 5. The concentrations of TPPU and regulatory lipid mediators involved in the CYP450 pathway in plasma.**
The graphs in panels A and B show the concentrations of TPPU and 14,15-DiHETrE, respectively, in plasma. All data are shown as the mean ± SEM (n = 6–8). The asterisks (*) denote significant ($p < 0.05$) differences compared to the PBS controls, while the brackets represent significant ($p < 0.05$) differences among the other three groups, according to a one-way ANOVA and post hoc Tukey's Multiple Comparisons Test. The heatmap in Panel C shows the higher concentrations of plasma EpFAs in the OVA + TPPU (6 h) group (yellow frame) and plasma diols in the OVA group (white frame. Abbreviations: DiHDPE—dihydroxydocosapentaenoic acid; DiHETE—dihydroxyeicosatetraenoic acid; DiHETrE—dihydroxyeicosatrienoic acid; DiHODE—dihydroxyoctadecadienoic acid; DiHOME— dihydroxyoctadecamonoenoic acid; EET—epoxyeicosatrienoic acid; EpDPE—epoxydocosapentaenoic acid; EpETE— epoxyeicosatetraenoic acid; EpFA—epoxy fatty acid; EpODE—epoxyoctadecadienoic acid; and EpOME— epoxyoctamonoemoic acid.

neutrophils) in a lipopolysaccharide-induced acute lung injury model [37] when compared to controls without TPPU.

**Effects of TPPU on COX/LOX-related BALF lipid mediators.** Lipid mediator concentrations in BALF supernatant (Fig 6A) showed a range of inflammatory lipid mediators— including 11,12-DiHETrE, 5-hydroxyeicosatetraenoic acid (HETE), LTB4, 9-hydroxyoctadecadienoic acid (HODE), and prostaglandin (PG) E2—were ≥ 2-fold lower in the OVA + TPPU groups relative to the OVA group. Their average concentrations (mean ± SEM), fold changes (FC) in the OVA + TPPU (2 h) versus OVA group, and corresponding $p$-values were shown in the S3 Table. Some with an FC ≤ 1/2 and a $p$-value < 0.05 were labeled as red dots in a volcano plot (Fig 6B) or shown as bar graphs (Fig 6C).

In previous publications, sEHIs modulated the concentrations of products in CYP450-adjacent pathways. Results of lipid analyses in the present study support that in addition to direct

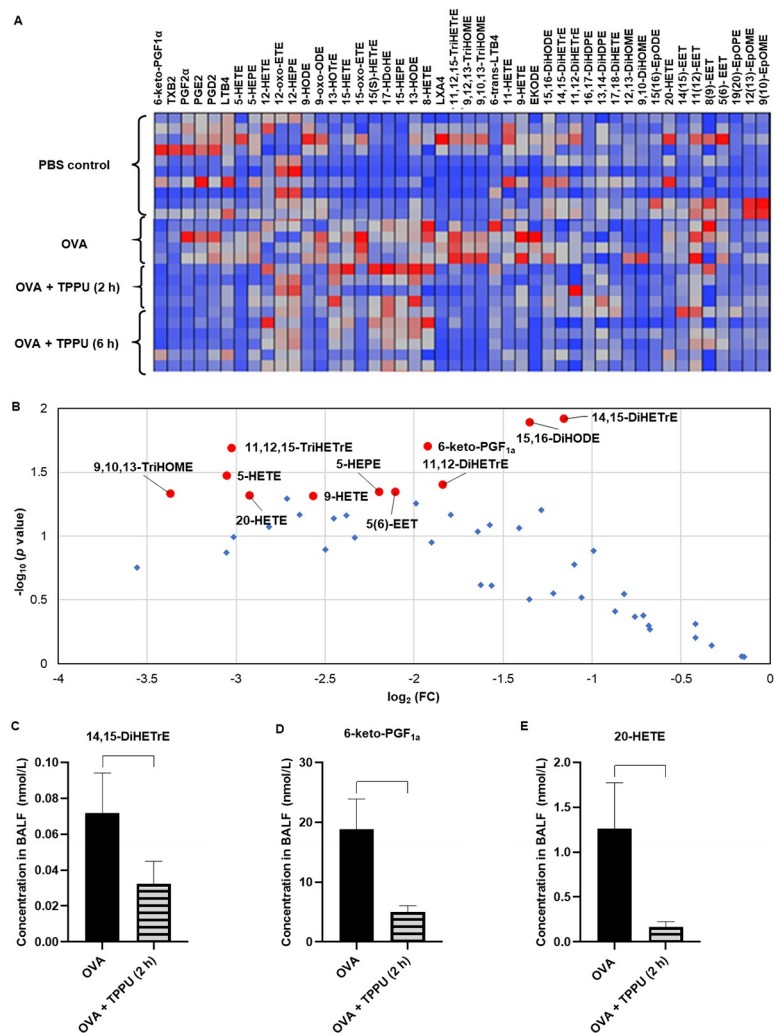

**Fig 6. Lipidomic analyses of BALF.** Panel A shows a heatmap of the lipidomic profiles involved in CYP450, COX, and LOX pathways from each exposure group. Panel B shows a volcano plot comparing lipids in the OVA and OVA + TPPU (2 h) treatment groups. Each dot represents a lipid mediator measured in the latter group, with log2 (FC) on the x-axis representing concentration changes relative to the OVA-only group and -log10 ($p$-value) on the y-axis representing the statistical significance of their difference. Among them, red dots in the plot represent lipid mediators with concentrations in the OVA + TPPU (2 h) group that were changed significantly ($p < 0.05$ in a one-way ANOVA), with a minimal 2-fold change, relative to the OVA group. The blue dots represent other lipid mediators, with concentrations in the OVA + TPPU (2 h) group that were not changed statistically relative to the OVA-only group. Panels C–E show box plots of the representative lipid mediators via the three pathways, with the most dramatic changes in the OVA + TPPU (2 h) versus OVA group. All data are shown as group means ± SEMs (n = 6–8). The brackets represent significant ($p < 0.05$) differences between the OVA and OVA + TPPU (2 h) groups, using a one-way ANOVA and post hoc Tukey's Multiple Comparisons Test. Abbreviations: DiHDPE—dihydroxydocosapentaenoic acid; DiHETE—dihydroxyeicosatetraenoic acid; DiHETrE—dihydroxyeicosatrienoic acid; DiHODE—dihydroxyoctadecadienoic acid; DiHOME—dihydroxyoctadecamonoenoic acid; EET—epoxyeicosatrienoic acid; EKODE—epoxyketooctadecenoic acid; EpDPE—epoxydocosapentaenoic acid; EpETE—epoxyeicosatetraenoic acid; EpODE—epoxyoctadecadienoic acid; EpOME—epoxyoctamonoemoic acid; ETE—eicosatrienoic acid; HDoHE—hydroxydocosahexaenoic acid; HETE—hydroxyeicosatetraenoic acid; HETrE—hydroxyeicosatrienoic acid; HEPE—hydroxyeicosapentanoic acid; HODE—hydroxyoctadecadienoic acid; HOME—hydroxyoctadecamonoenoic acid; HOTrE—hydroxyoctadecatrienoic acid; LX—lipoxin; LT—leukotriene; PG—prostaglandin; and TX—thromboxane.

inhibition of sEHs, TPPU produces indirect anti-inflammatory impacts through the COX and LOX pathways (Fig 6).

First, pro-inflammatory lipid mediators via the LOX pathway (i.e., LTs) were decreased following OVA+TPPU exposure compared to OVA alone. This finding reflected a reduction in bronchoconstriction [31]. Studies of a different sEHI, trans-4-{4-[3-(4-trifluoromethoxyphenyl)-ureido] cyclohexyloxy} benzoic acid (*t*-TUCB), yielded similar results to the present study on TPPU in preventing the influx of inflammatory cells into lung tissues and BALF of asthmatic mice. *T*-TUCB also 1) increased a few anti-inflammatory COX and LOX metabolites (e.g., 17-hydroxydocosahexaenoic acid (HDoHE)) in the plasma and BALF while decreasing the pro-inflammatory mediators (e.g., 6-keto-PGF1α and LTB4) [24]; and 2) attenuated eosinophil recruitment in a murine model of food allergy [58]. These *t*-TUCB findings may explain the mechanism by which TPPU produces anti-inflammatory effects in the lungs.

The concomitant inhibition of elevated mucosubstances in the airways and PG, HETE, and LTB4 formation observed with TPPU administration in the present study are also supported by previous studies of *t*-TUCB. Dramatic reductions in T-helper 2 (Th2) cytokines were noted with [24] decreases in intraepithelial mucosubstances after *t*-TUCB administration in an OVA-induced asthmatic murine model. Th2 cytokines promote mucus hypersecretion and asthma pathogenesis [48, 49] by affecting eicosanoid metabolism occurring in human bronchial epithelial cells mainly via the COX and LOX pathways. The pathways typically lead to releases of PGs and 15-HETE (S1 Fig). PG deficiencies in airways after TPPU treatment may improve airway remodeling and chronic inflammation [49]. Likewise, a reduction in LTB4 has been shown to alleviate vascular permeability and mucus production in a murine asthma model [31]. The increased LXs observed with TPPU exposure in the present study suggest that LXs might also contribute to the benefits of sEHI administration. The cumulative findings led us to speculate that inhaled TPPU may influence bronchial epithelial cell PUFA metabolism in the COX and LOX pathways by suppressing Th2 cytokine expression and reducing the relevant pro-inflammatory lipid mediators to limit airway remodeling.

Other mechanisms not explored in the present study but reported for other sEHIs included downregulation of c-Jun N-terminal kinases (critical regulators involved in the processes of pro-inflammation, tissue remodeling, and apoptosis) and other cytokines [16]; reduced expression of remodeling-related markers in BALF and lung tissues; and decreased and collagen deposition in lungs [17]. Though, in previous studies, airway remodeling was frequently related and not always second to airway inflammation [59], results from the present study suggested sEHIs may effectively alleviate both airway remodeling and inflammation.

## Conclusions

When considered as a whole, the findings of the present preliminary study indicate the TPPU-exposed groups benefited from indirect and downstream impacts [easing inflammation (Fig 4), AHR (Figs 2, 3A and 3B), and remodeling (Fig 3A and 3C)] through the anti-inflammatory action of several EpFAs (e.g., EETs; Figs 5 and 6); the reduced formation of diols (e.g., DiHETrEs and DiHOMEs) in plasma and BALF (Figs 5 and 6); and the moderation of pro-inflammatory lipid mediators (e.g., 6-keto-PGF1α and 20-HETE) involved in the COX and LOX pathways (Fig 6).

The route of TPPU administration is likely to influence its effects on asthma and asthma symptoms. In previous studies, other sEHIs were administrated subcutaneously [24], orally [16, 28], or intragastrically [17, 27] to attenuate airway allergic inflammation [24, 27] and hyper-responsiveness [24] in murine models, improve lung function and attenuate emphysema in a rat model of chronic obstructive pulmonary disease caused by tobacco smoke

exposure [28], or reduce airway remodeling [16] and hyper-responsiveness [16] in a mouse model of chronic asthma. As TPPU inhalation studies are not found in the current literature, the present study is the first report on the efficacy of inhaled TPPU, or inhaled sEHIs in general, to mitigate asthma disease outcomes in murine models. Although administration of TPPU by IP [37] or intragastric injection [29] may have potential therapeutic effects on lung disease, delivery of TPPU directly to the pulmonary system via inhalation is advantageous in that it 1) offers immediate, convenient, and direct delivery of the treatment to the targeted tissues of the lungs; 2) requires lower treatment doses relative to systemic administration with distribution to non-targeted tissues; and 3) reduces the potential for adverse health impacts due to the generation of toxic metabolites with systemic administration. The use of nebulizers is the preferred route of drug delivery for pulmonary diseases [60]. Hence, our confirmed hypothesis that inhaled TPPU prevents exacerbation of asthma or other respiratory diseases by reducing airway inflammation, hyper-responsiveness, and remodeling is notable for future studies to determine the pharmacokinetics (i.e., absorption, distribution, metabolism, and excretion) and pharmacodynamics (i.e., concentration/dose responses) of inhaled TPPU, explore its preventive or therapeutic efficacy and potential as an asthma preventative or rescue treatment, and progress toward future human clinical trials.

Study limitations included but were not limited to the small sample size that limited data points for different endpoints (e.g., PFTs), missing lung weight measurements, and exclusion of female mice. Additional experiments designed to expand upon our findings should account for the study limitations and 1) include long-term experiments with TPPU since chronic asthma leads to chronic pulmonary changes, such as fibrosis; 2) quantify asthma-related cytokines (e.g., IL-4, -15, and -13) and bronchoalveolar lavage cells (e.g., via total counts, viable/non-viable ratios, and differentials) to support conclusions derived from histopathological findings; 3) perform inter-group comparisons of absolute and relative (to body weight) lung weights; 4) test male and female mice of various ages to allow for comparisons of the sex-related impacts of TPPU at different life stages; and 5) examine the effects of concurrent exposures to hydroxyeicosatrienoic acids (HETrEs) or LTs and TPPU to determine the strength and range of TPPU's effects on AHR and inflammation in the presence/absence of increased (exogenous) pro-inflammatory lipid metabolite concentrations.

Overall, our results indicated aerosolized inhalable TPPU is a promising novel candidate for the preventative and acute treatment of asthma due to its anti-inflammatory effects, inhibition of AHR, and attenuation of airway epithelial remodeling. Including asthmatic models exposed to other, more physiologically relevant allergens (e.g., pollen, house dust mites) in future studies should provide additional persuasive results to support clinical studies on the therapeutic use of TPPU as a potential asthma treatment.

## Supporting information

**S1 Table. Semi-quantitative scoring rubric for the severity of inflammation in lung tissues.** (DOCX)

**S2 Table. Semi-quantitative scoring rubric for the severity of airway constriction.** (DOCX)

**S3 Table. Concentrations of lipid mediators in BALF of mice inhaling OVA + TPPU (2 h) versus OVA alone.** (DOCX)

**S1 Fig. The main metabolic pathways of PUFAs.** These pathways combine to produce complex products. The CYP450 pathway also produces a variety of allylic hydroxy metabolites and

omega (ω) and ω-1 hydroxy metabolites, such as the pro-inflammatory vasoconstrictor 20-HETE (hydroxyeicosatetraenoic acid). Abbreviations: COX—cyclooxygenase; CYP450—cytochrome P450; HDPEs—hydroxydocosapentaenoic acids; HEPEs—hydroxyeicosapentanoic acids; HETrEs—hydroxyeicosatrienoic acids; LOX—lipoxygenase; PUFA—polyunsaturated fatty acid; sEH—soluble epoxide hydrolase.
(TIF)

**S2 Fig. Semi-quantitative histopathological scoring of perivascular and pleural inflammation in mice.** Data are shown as the mean ± standard error of the mean. An asterisk (*) denotes a significant ($p < 0.05$) difference compared to the PBS control, while brackets represent significant ($p < 0.05$) differences between the OVA and OVA + TPPU groups. A Kruskal-Wallis test and post-hoc Dunnett's Multiple Comparisons test were used to analyze perivascular inflammation. A one-way ANOVA and post hoc Tukey's Multiple Comparisons test were used for pleural inflammation.
(TIF)

# Acknowledgments

The authors are grateful to the National Key R&D Program of China (2019YFE0117800); Christine Batista and Ciara Fulgar for assistance in biological sample analysis; Jasmine Singh for semi-quantitative scoring of lung inflammation; Dr. Rona M. Silva for constructive editorial advice on the manuscript.

# Author Contributions

**Formal analysis:** Debin Wan, Savannah Mack, Jingjing Zhang, Karen Wagner, Chang Wang, Bowen Tan, Jason Chen, Ching-Wen Wu, Kaori Tsuji, Minoru Takeuchi.

**Writing – original draft:** Chuanzhen Zhang, Wei Li, Xiyuan Li.

**Writing – review & editing:** Ziping Chen, Bruce D. Hammock, Kent E. Pinkerton, Jun Yang.

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
