## [Decision Letter · Decision Letter 0]

20 Jan 2022

PONE-D-21-40700Novel aerosol treatment of airway hyper-reactivity and inflammation in a murine model of asthma with a soluble epoxide hydrolase inhibitorPLOS ONE

Dear Dr. Wu,

Thank you for submitting your manuscript to PLOS ONE. After careful consideration, we feel that it has merit but does not fully meet PLOS ONE’s publication criteria as it currently stands. Therefore, we invite you to submit a revised version of the manuscript that addresses the points raised during the review process.

We look forward to receiving your revised manuscript.

Kind regards,

Saba Al Heialy

Academic Editor

PLOS ONE

Journal Requirements:

"Dr. Hammock and Dr. Yang are employed partially by EicOsis, which is developing a potent sEHI EC 5026 for pain relief. The other authors declared no competing interests."

We note that you received funding from a commercial source: EicOsis

3.Please include your full ethics statement in the ‘Methods’ section of your manuscript file. In your statement, please include the full name of the IRB or ethics committee who approved or waived your study, as well as whether or not you obtained informed written or verbal consent. If consent was waived for your study, please include this information in your statement as well. 

Reviewers' comments:

Reviewer's Responses to Questions

**Comments to the Author**

1. Is the manuscript technically sound, and do the data support the conclusions?

Reviewer #1: Yes

Reviewer #2: Partly

2. Has the statistical analysis been performed appropriately and rigorously? 

Reviewer #1: Yes

Reviewer #2: Yes

3. Have the authors made all data underlying the findings in their manuscript fully available?

Reviewer #1: Yes

Reviewer #2: Yes

4. Is the manuscript presented in an intelligible fashion and written in standard English?

Reviewer #1: Yes

Reviewer #2: Yes

5. Review Comments to the Author

Reviewer #1: The work by Zhang et al describes the results of testing an inhaled soluble epoxide hydrolase inhibitor TPPU (1-(4-trifluoro-methoxy-phenyl)-3-(1-propionylpiperidin-4-yl) urea) to prevent asthma by reducing airway inflammation, hyperresponsiveness and remodeling. The work is generally well done, and no additional experiments have to be added, but there are a few minor issues that need to be addressed before publication:

1) Line 97: Please clarify the role of lipoxines in asthma, are they good or bad, and what are consequences of increased lipoxines upon sEHI administration?

2) Line 110: LT is abbreviation for leukotrienes, I presume?

3) Line 115: Do we know if TPPU is binding to some other targets? How specific is this drug?

4) Line 146: Which device was used to deliver the inhaled TPPU?

Overall, this manuscript provides a large body of data that will be useful to guide further clinical trials. It also describes a first inhaled sEH inhibitor in murine asthma model. There are some missing parts, such as the effects of TPPU on lung weights. However, I recommend the manuscript for publication, after the authors fix the minor modifications noted above.

Reviewer #2: This study presents the results of aerosolized administration of a soluble epoxide hydrolase inhibitor (sEHI) 1‐trifluoromethoxyphenyl-3-(1-propionylpiperidin-4-

yl) urea, TPPU) to mice in a model of airway inflammation and airway hyper-responsiveness (AHR). Previously these authors demonstrated that subcutaneous administration of a similar sEHI reduced OVA-induced airway inflammation and AHR (ref 26).

Here they found that concurrent inhalational challenge with OVA and TTPU reduced allergic airway inflammation, AHR, and levels of some lipid mediators in the airways. An analog of TPPU EC5026 has already been used in human subjects as an oral analgesic, supporting the feasibility of this therapeutic approach. Although the paper is well written and the data mostly support the conclusions, the asthma model used here is outdated, and there are several gaps in data presentation and interpretation that need to be addressed to solidify these preliminary findings.

Major comments:

1. The major limitation of the study is the OVA-induced model of AHR is insufficient to justify the therapeutic use of sEHI. Parallel studies using a physiologically relevant allergen such as pollen, house dust mite, or fungus should need to be shown.

2. Why are only male mice used? Are there sex-based differences in EpFAs, EETs etc.?

3. As the authors note, plasma levels of TTPU were extremely high (80-140x above the IC50), and, as they also note, pulmonary concentrations are likely much higher. What is the justification for the concentration and duration of inhalation used? Since these high concentrations nearly obliterated inflammatory responses, it would be necessary to perform dose response studies of TTPU to demonstrate the specificity of the response.

4. Fig. 3B The assessment of static bronchoconstriction in fixed lung sections is of unclear significance. Please provide references to justify this approach.

5. Fig. 4 BALF total cell counts and composition, as well as levels of asthma-related cytokine (eg IL-4, 5, 13) are needed to support the conclusions derived from histopathological findings.

6. Figs 6, Supp tables The major differences in BALF lipid metabolites in OVA-challenged mice induced by TTPU appear to be in the bronchoconstrictive LOX-pathway. Does concurrent administration of HETrEs or LTs with TTPU reduce or reverse its ability to modulate AHR and/or inflammation in this model?

Minor comments:

-Intro, line 66 discussion of treatment options completely ignores recent success of biologics

targeting asthma-related cytokines

-Discussion, line 444 Do you mean weight loss?

-Discussion contains much re-statement of results that provide no additional insights and should be shortened.

6. PLOS authors have the option to publish the peer review history of their article (what does this mean?). If published, this will include your full peer review and any attached files.

Reviewer #1: No

Reviewer #2: No

---

## [Author Response · Author response to Decision Letter 0]

14 Mar 2022

March 5, 2022

Saba Al Heialy

Academic Editor

PLoS One

RE: PONE-D-21-40700, “Novel aerosol treatment of airway hyper-reactivity and inflammation in a murine model of asthma with a soluble epoxide hydrolase inhibitor”

Dear Dr. Saba Al Heialy:

Thank you for reviewing our manuscript listed above and its merit for publication in PLoS One. We are pleased to submit our revised manuscript addressing the points raised during the review process.

Please find our replies to each comment raised by the editor and reviewers accompanying this letter. We have uploaded marked and unmarked copies of our revised manuscript as part of the resubmission process. The replies include page and line numbers corresponding to the unmarked copy of our revised manuscript. The revisions are highlighted in the marked copy labeled “Revised Manuscript with Track Changes,” while the final clean version is labeled “Manuscript.” 

We have no restrictions on our adherence to PLoS One’s policies on sharing the data and materials in our revised manuscript. Thank you for your further consideration of our revised manuscript, which we feel has been significantly improved with greater clarity. 

Sincerely,

Chuanzhen Zhang, M.D. 

Physician/Scientist

Kent E. Pinkerton, Ph.D.

Professor

Corresponding Author Contact Information

Jun Yang, Ph.D.

Department of Entomology and Nematology and Comprehensive Cancer Center

University of California, Davis

One Shields Avenue

Davis, CA 95616, USA. 

Email: junyang@ucdavis.edu

Phone: +1 (530) 752 5109

Fax: +1 (530) 752 1537

Wei Li, Ph.D.

School of Control Science and Engineering

Shandong University

16766 Jingshi Road, Jinan, 250014, China

Email: cindy@sdu.edu.cn

Tel: +86 531 8839 2625

FAX: +86 531 8839 2205

Kent E. Pinkerton, Ph.D.

Center for Health and the Environment

University of California, Davis

One Shields Avenue

Davis, CA 95616, USA

Email: kepinkerton@ucdavis.edu

Phone: +1 (530) 752-8334

Fax: +1 (530) 752-5300 

Comments from the Editor and Other Reviewers

Editor’s Comments

Comment 1: If applicable, we recommend that you deposit your laboratory protocols in protocols.io to enhance the reproducibility of your results. Protocols.io assigns your protocol its own identifier (DOI) so that it can be cited independently in the future. For instructions see: https://journals.plos.org/plosone/s/submission-guidelines#loc-laboratory-protocols. Additionally, PLOS ONE offers an option for publishing peer-reviewed Lab Protocol articles, which describe protocols hosted on protocols.io. Read more information on sharing protocols at https://plos.org/protocols?utm_medium=editorial-email&utm_source=authorletters&utm_campaign=protocols.

REPLY 1: We thank the editor for this suggestion. The protocol for developing the murine OVA-induced asthma model was previously published (Yang et al. 2015) and cited in our originally submitted manuscript. The citation appears on page 9, line 150 of the revised document. While the TPPU inhalation exposure protocol is new, we provided a description in the Materials and Methods section of the revised manuscript (page 9, lines 157–162). 

Comment 2: Journal Requirements:

REPLY 2: We have made changes to ensure our manuscript meets the journal’s style and file naming requirements. 

Comment 3A: Thank you for stating the following in the Competing Interests section: 

"Dr. Hammock and Dr. Yang are employed partially by EicOsis, which is developing a potent sEHI EC 5026 for pain relief. The other authors declared no competing interests."

We note that you received funding from a commercial source: EicOsis

REPLY 3A: No funding was received from EicOsis for this study. This was clarified in the Competing Interests section of our revised manuscript (page 42, lines 843–850).

Comment 3B: Please provide an amended Competing Interests Statement that explicitly states this commercial funder, along with any other relevant declarations relating to employment, consultancy, patents, products in development, marketed products, etc. 

REPLY 3B: Modifications have been made to the Competing Interests section (page 42, lines 843–850) of our revised manuscript, as shown below. In the accompanying cover letter, we reiterate that we have no restrictions on our adherence to PLoS One’s policies on sharing data and materials from the present study.

“B.D. Hammock and J. Yang are co-authors on University of California patents for epoxide hydrolase inhibitors (sEHIs). An sEHI which is structurally similar to the TPPU used in this study, is being developed by EicOsis Human Health to replace opioid analgesics in treating pain. Drs. Hammock and Yang are part-time employees of EicOsis Human Health. This does not alter our adherence to PLoS One policies on sharing data and materials. Data in this publication were not generated from grant- or salary-money from EicOsis. This research has no financial connections to EicOsis. No confidential University information has been provided to EicOsis by the University of California or its employees.”

Comment 4: Please include your full ethics statement in the ‘Methods’ section of your manuscript file. In your statement, please include the full name of the IRB or ethics committee who approved or waived your study, as well as whether or not you obtained informed written or verbal consent. If consent was waived for your study, please include this information in your statement as well. 

REPLY 4: The Materials and Methods section (page 8, lines 132–134) of our manuscript has been revised to state the following.

“All animal experiments were approved in writing by the University of California Davis’ Institutional Animal Care and Use Committee (IACUC) to ensure the proper care and treatment of animals in our research.” 

 

Reviewer #1 Comments 

Comment 1: The work by Zhang et al describes the results of testing an inhaled soluble epoxide hydrolase inhibitor TPPU (1-(4-trifluoro-methoxy-phenyl)-3-(1-propionylpiperidin-4-yl) urea) to prevent asthma by reducing airway inflammation, hyperresponsiveness and remodeling. The work is generally well done, and no additional experiments have to be added, but there are a few minor issues that need to be addressed before publication:

1) Line 97: Please clarify the role of lipoxins in asthma, are they good or bad, and what are consequences of increased lipoxins upon sEHI administration?

2) Line 110: LT is abbreviation for leukotrienes, I presume?

3) Line 115: Do we know if TPPU is binding to some other targets? How specific is this drug?

4) Line 146: Which device was used to deliver the inhaled TPPU?

REPLY 1: We thank the reviewer for the opportunity to clarify our manuscript and have included numbered responses corresponding to the reviewer’s queries. 

1) Lipoxins (LXs) mitigate the adverse effects of asthma and increase with sEHI administration. The revised manuscript provides the following details (page 6, lines 92–99, and page 26, lines 518–520, respectively) as shown below.

“Asthma severity has been linked to imbalances in the levels of LTs relative to LXs, both of which are products of lipoxygenase-derived eicosanoid biosynthesis. For example, higher mean levels of pro-inflammatory LTs and lower basal levels of LXs have been reported in patients with severe versus moderate asthma [Levy et al., 2005]. Several studies show that LXs are generally beneficial, promoting the resolution of allergic inflammation [Karra et al., 2015]. In a separate study [Ono et al., 2014], we showed that LX generation is related to sEH activity in severe asthmatic patients. LXs may regulate allergic airway responses (e.g., inflammation, mucus metaplasia, and AHR) by inhibiting eosinophil trafficking, neutrophil chemotaxis, and degranulation [Ono et al., 2014].”

“The increased LXs observed with TPPU exposure in the present study suggest that LXs might contribute to the benefits of sEHI administration.”

2) Yes, “LT” is the abbreviation for “leukotriene,” as indicated in the revised manuscript (page 4, line 59).

3) No reports have confirmed that TPPU directly binds to non-sEH targets. However, TPPU has been used in studies of other morbidities, such as hypertension (Bukhari et al., 2020). For example, in the study by Bukhari et al., TPPU appeared to moderate hypertension by inhibiting angiotensin-converting enzyme (ACE) gene expression. The mechanism of TPPU’s action on ACE was not elucidated in the study. However, ACE is a critical enzyme in the renin-angiotensin system that controls blood pressure by regulating the volume of fluids in the body. ACE converts the angiotensin I hormone to angiotensin II, an active vasoconstrictor. 

Thus far, sEH is the only TPPU target we have identified. Separate from the present study, we have screened TPPU against functional hERG (the human Ether-a-go-go-Related Gene) that codes for a protein known as Kv11.1, the alpha subunit of a potassium ion channel, using CEREP profile tests. These tests enable the early identification of significant off-target interactions. The results to date have all been negative.

We have added the following comment in the revised manuscript (page 7, line 117). 

“TPPU has not been shown to bind to other (non-sEH) targets.”

4) The TPPU aerosol was generated using a Mini-Heart Lo-Flo Nebulizer (Westmed Inc., Tucson, AZ) attached to a nose-only exposure chamber. This statement was added to the revised manuscript (page 9, lines 157–158).

Comment 2: Overall, this manuscript provides a large body of data that will be useful to guide further clinical trials. It also describes a first inhaled sEH inhibitor in murine asthma model. There are some missing parts, such as the effects of TPPU on lung weights. However, I recommend the manuscript for publication, after the authors fix the minor modifications noted above.

REPLY 2: We thank the reviewer for the comment that our manuscript contains data that will be useful to guide further clinical trials. We also appreciated the reviewer’s recommendation for our manuscript to be published in PLoS One.

The procedural nature of this study, with in situ bronchoalveolar lavage (BAL) and inflation-fixation of lung tissues, precluded lung weights from being taken. We have added a statement (page 28, lines 567–568 of the revised manuscript) to indicate that future studies should consider measuring lung weights upon necropsy to allow inter-group comparisons. 

Reviewer #2 Comments

This study presents the results of aerosolized administration of a soluble epoxide hydrolase inhibitor (sEHI) 1‐trifluoromethoxyphenyl-3-(1-propionylpiperidin-4-yl) urea, TPPU) to mice in a model of airway inflammation and airway hyper-responsiveness (AHR). Previously these authors demonstrated that subcutaneous administration of a similar sEHI reduced OVA-induced airway inflammation and AHR (ref 26).

Here they found that concurrent inhalational challenge with OVA and TTPU reduced allergic airway inflammation, AHR, and levels of some lipid mediators in the airways. An analog of TPPU EC5026 has already been used in human subjects as an oral analgesic, supporting the feasibility of this therapeutic approach. Although the paper is well written and the data mostly support the conclusions, the asthma model used here is outdated, and there are several gaps in data presentation and interpretation that need to be addressed to solidify these preliminary findings.

Major Comments

Comment 1: The major limitation of the study is the OVA-induced model of AHR is insufficient to justify the therapeutic use of sEHI. Parallel studies using a physiologically relevant allergen such as pollen, house dust mite, or fungus should need to be shown.

REPLY 1: We agree with the reviewer that other relevant allergen (e.g., pollen, house dust mite (HDM), and/or fungus)-induced asthma models would be advantageous to test the efficacy of TPPU treatment further. The inclusion of such models is beyond the scope of the present preliminary study, but we have recommended it in the revised manuscript (page 28, lines 576–579) for future studies. 

We hope to develop HDM- and fungus-induced asthma animal models to verify the therapeutic effects of soluble epoxide hydrolase inhibitors (sEHIs) in the future. However, we think the findings from the present study merit publication given the reviewer-directed changes that have enhanced the quality and clarity of our revised manuscript.

In full disclosure, we thought it would be prudent to use an OVA-induced asthma model to maintain consistency with our previous study (Yang et al., 2015), which also used an OVA-induced asthma model. In the current study, we tested a different sEHI, i.e., 1‐trifluoromethoxyphenyl-3-(1-propionylpiperidin-4-yl) urea (TPPU) rather than trans-4-{4-[3-(4-trifluoromethoxyphenyl)-ureido] cyclohexyloxy} benzoic acid (t-TUCB). We also used inhalation as a novel TPPU administration method. The OVA-induced asthma model is a classic acute asthma murine model (Debeuf et al., 2016) that remains relevant for current airway inflammation studies (Tang et al., 2021). In the present study, the OVA-induced murine model produced airway inflammation, hyper-responsiveness, and remodeling, which allowed comparisons between the OVA-induced asthma group, the non-sensitized (PBS-exposed) negative control group, and the two TPPU+OVA treatment groups. 

Comment 2: Why are only male mice used? Are there sex-based differences in EpFAs, EETs etc.?

REPLY 2: Previous studies have reported sex-related differences in soluble epoxide hydrolase (sEH) expression and activity in various tissues, often with estrogen-dependent downregulation of sEH potentiating epoxyeicosatrienoic acid (EET) bioavailability and an anti-inflammatory state in female versus male mice (Qin et al., 2015; Huang et al., 2018; Yang et al., 2018; Grant et al., 2019). Given the preliminary nature of the present study and our limited resources, we did not study sex dimorphisms. Instead, we focused on male mice since they have a higher circulating target (sEH) enzyme concentration than female mice. However, we agree that both sexes should be studied and have added a recommendation (page 28, lines 568–570) for future studies to compare sex-related impacts of inhaled TPPU at various life stages. 

Comment 3: As the authors note, plasma levels of TTPU were extremely high (80-140x above the IC50), and, as they also note, pulmonary concentrations are likely much higher. What is the justification for the concentration and duration of inhalation used? Since these high concentrations nearly obliterated inflammatory responses, it would be necessary to perform dose response studies of TTPU to demonstrate the specificity of the response.

REPLY 3: We agree with the reviewer. One goal of the present preliminary study was to verify whether inhalation of TPPU would work as well as subcutaneous administration of the sEHI (t-TUCB) did in our previous study (Yang et al., 2015). Dose-response experiments were performed in the study of t-TUCB, and the dosage of TPPU in the present study was optimized to match the plasma level of TPPU to that of t-TUCB in our previous study. 

Our original manuscript stated that the pharmacokinetics (PK) and pharmacodynamics (PD) of TPPU should be explored. Dose-response is part of PD, so we revised the text to state that future studies should determine the “pharmacokinetics (i.e., absorption, distribution, metabolism, and excretion) and pharmacodynamics (i.e., concentration/dose responses)” of inhaled TPPU (page 2, lines 38–40; page 27, lines 556–558). 

Comment 4: Fig. 3B The assessment of static bronchoconstriction in fixed lung sections is of unclear significance. Please provide references to justify this approach.

REPLY 4: The approach taken to measure static bronchoconstriction in the present study was rather novel. However, we felt it could help further explain and morphologically demonstrate our active physiological measurements of airway resistance. This novel approach was developed by a summer high school student (Jason Chen) working in the laboratory. Using a computerized planimetry approach with ImageJ software in the public domain, a pre-defined ordinal scale for the severity of airway constriction (shown in supplemental Table 2), and the ratio of constricted airways to total airways (constricted airways/total airways), an airway constriction score was calculated for each lung level. A total of four lung levels were scored and averaged to assign an overall airway constriction score for each mouse.

Comment 5: Fig. 4 BALF total cell counts and composition, as well as levels of asthma-related cytokine (eg IL-4, 5, 13) are needed to support the conclusions derived from histopathological findings.

REPLY 5: Given the preliminary nature of our study and limited resources, we could not measure as many biological outcomes as we hoped. Our primary objectives were to 1) use a novel therapeutic, TPPU, by inhalation; 2) determine its efficacy in reducing airway hyper-responsiveness (AHR) and inflammation and ameliorating asthmatic-like symptoms; and 3) compare lipidomic profiles of mice exposed and not exposed to TPPU. We agree quantifying asthma-related cytokine concentrations would be good. Thus, in the revised manuscript (page 28, lines 567–567), we recommend future studies “quantify asthma-related cytokines” such as IL-4, IL-15, and IL-13.

In a previous study by our group (Yang et al.,2015), t-TUCB administration dramatically decreased IL-4, IL-5, and eotaxin, an eosinophil-stimulating chemokine. These effects of t-TUCB on Th2 cytokines are mentioned on page 25 (lines 511–513) of the revised manuscript.

Comment 6: Figs 6, Supp tables The major differences in BALF lipid metabolites in OVA-challenged mice induced by TTPU appear to be in the bronchoconstrictive LOX-pathway. Does concurrent administration of HETrEs or LTs with TTPU reduce or reverse its ability to modulate AHR and/or inflammation in this model?

REPLY 6: The question is intriguing. We have not tested whether combined hydroxy-eicosatrienoic acid (HETrE) or leukotriene (LT) administration with TPPU alters its ability to modulate airway hyper-responsiveness (AHR) or inflammation. Thus, on page 28, lines 570–573 of the revised manuscript, we suggested that future studies “examine the effects of concurrent exposures to hydroxyeicosatrienoic acids (HETrEs) or LTs and TPPU to determine the strength and range of TPPU’s effects on AHR and inflammation in the presence/absence of increased (exogenous) pro-inflammatory lipid metabolite concentrations.”

Minor Comments

Comment 7: Intro, line 66 discussion of treatment options completely ignores recent success of biologics targeting asthma-related cytokines

REPLY 7: We agree advances in biologics targeting asthma-related cytokines are noteworthy. Thus, we have added the following text to the revised manuscript (page 4, lines 61–64), as shown below. 

“New biologics have recently been developed to target effector cells, cytokines, and molecules to inhibit eosinophils, interleukin (IL)-5, IL-9, immunoglobulin E, and epithelial-derived cytokines involved in asthma [Hammad et al., 2021]. These novel compounds have proven efficacy in targeting specific drivers of asthma-related symptoms.” 

Comment 8: Discussion, line 444 Do you mean weight loss?

REPLY 8: Yes. The text in the revised document (page 16, lines 317–318) has been changed to state, “Future studies should confirm whether weight loss is a side-effect of TPPU by measuring body weights throughout the study.”

Comment 9: Discussion contains much re-statement of results that provide no additional insights and should be shortened.

REPLY 9: The Results and Discussion sections have been combined for conciseness. 

REFERENCES

Bonnans C, Chanez P, Chavis C. Lipoxins in asthma: potential therapeutic mediators on bronchial inflammation? Allergy. 2004; 59(10):1027-1041. doi:10.1111/j.1398-9995.2004.00617.x

Bukhari IA, Alorainey BI, Al-Motrefi AA, Mahmoud A, Campbell WB, Hammock BD. 1-trifluoromethoxyphenyl-3-(1-propionylpiperidin-4-yl) urea (TPPU), a soluble epoxide hydrolase inhibitor, lowers L-NAME-induced hypertension through suppression of angiotensin-converting enzyme in rats. Eur Rev Med Pharmacol Sci. 2020;24(15):8143-8150. doi:10.26355/eurrev_202008_22501

Celik G, Misirligil Z. Lipoxins in asthma. J Allergy Clin Immunol. 2004;114(4):992. doi:10.1016/j.jaci.2004.06.012

Cui Z, Li B, Zhang Y, He J, Shi X, Wang H, Zhao Y, Yao L, Ai D, Zhang X, Zhu Y. Inhibition of Soluble Epoxide Hydrolase Attenuates Bosutinib-Induced Blood Pressure Elevation. Hypertension. 2021; 78(5):1527-1540. doi: 10.1161/HYPERTENSIONAHA.121.17548

Debeuf N, Haspeslagh E, van Helden M, Hammad H, Lambrecht BN. Mouse Models of Asthma. Curr Protoc Mouse Biol. 2016; 6(2):169-184. Published 2016 Jun 1. doi:10.1002/cpmo.4

Hammad H, Lambrecht BN. The basic immunology of asthma. Cell. 2021; 184(9):2521-2522. doi: 10.1016/j.cell.2021.04.019. Erratum for: Cell. 2021; 184(6):1469-1485

Huang A, Sun D. Sexually Dimorphic Regulation of EET Synthesis and Metabolism: Roles of Estrogen. Front Pharmacol. 2018; 9:1222. Published 2018 Oct 29. doi:10.3389/fphar.2018.01222

Grant MKO, Seelig DM, Sharkey LC, Choi WSV, Abdelgawad IY, Zordoky BN. Sexual dimorphism of acute doxorubicin-induced nephrotoxicity in C57Bl/6 mice. PLoS One. 2019;14(2):e0212486. Published 2019 Feb 20. doi:10.1371/journal.pone.0212486

Karra L, Haworth O, Priluck R, Levy BD, Levi-Schaffer F. Lipoxin B₄ promotes the resolution of allergic inflammation in the upper and lower airways of mice. Mucosal Immunol. 2015; 8(4):852-862. doi:10.1038/mi.2014.116

Levy BD, Bonnans C, Silverman ES, Palmer LJ, Marigowda G, Israel E. Severe Asthma Research Program, National Heart, Lung, and Blood Institute. Diminished lipoxin biosynthesis in severe asthma. Am J Respir Crit Care Med. 2005; 172(7):824-30. doi: 10.1164/rccm.200410-1413OC

Ono E, Dutile S, Kazani S, et al. Lipoxin generation is related to soluble epoxide hydrolase activity in severe asthma. Am J Respir Crit Care Med. 2014; 190(8):886-897. doi:10.1164/rccm.201403-0544OC 

Qin J, Kandhi S, Froogh G, et al. Sexually dimorphic phenotype of arteriolar responsiveness to shear stress in soluble epoxide hydrolase-knockout mice. Am J Physiol Heart Circ Physiol. 2015; 309(11): H1860-H1866. doi:10.1152/ajpheart.00568.2015

Tang W, Dong M, Teng F, et al. TMT-based quantitative proteomics reveals suppression of SLC3A2 and ATP1A3 expression contributes to the inhibitory role of acupuncture on airway inflammation in an OVA-induced mouse asthma model. Biomed Pharmacother. 2021; 134:111001. doi:10.1016/j.biopha.2020.111001

Yang J, Bratt J, Franzi L, et al. Soluble epoxide hydrolase inhibitor attenuates inflammation and airway hyperresponsiveness in mice. Am J Respir Cell Mol Biol. 2015; 52(1):46-55. doi:10.1165/rcmb.2013-0440OC

Yang YM, Sun D, Kandhi S, et al. Estrogen-dependent epigenetic regulation of soluble epoxide hydrolase via DNA methylation. Proc Natl Acad Sci USA. 2018; 115(3):613-618. doi:10.1073/pnas.1716016115

---

## [Editor Report · Decision Letter 1]

24 Mar 2022

Novel aerosol treatment of airway hyper-reactivity and inflammation in a murine model of asthma with a soluble epoxide hydrolase inhibitor

PONE-D-21-40700R1

Dear Dr. Wu,

We’re pleased to inform you that your manuscript has been judged scientifically suitable for publication and will be formally accepted for publication once it meets all outstanding technical requirements.

Kind regards,

Saba Al Heialy

Academic Editor

PLOS ONE
---

## [Editor Report · Acceptance letter]

5 Apr 2022

PONE-D-21-40700R1 

Novel aerosol treatment of airway hyper-reactivity and inflammation in a murine model of asthma with a soluble epoxide hydrolase inhibitor 

Dear Dr. Wu:

I'm pleased to inform you that your manuscript has been deemed suitable for publication in PLOS ONE. Congratulations! Your manuscript is now with our production department. 

Kind regards, 

on behalf of

Dr. Saba Al Heialy 

Academic Editor

PLOS ONE